# Metagenomic of Liver Tissue Identified at Least Two Genera of Totivirus-like Viruses in *Molossus molossus* Bats

**DOI:** 10.3390/microorganisms12010206

**Published:** 2024-01-19

**Authors:** Roseane da Silva Couto, Endrya do Socorro Foro Ramos, Wandercleyson Uchôa Abreu, Luis Reginaldo Ribeiro Rodrigues, Luis Fernando Marinho, Vanessa dos Santos Morais, Fabiola Villanova, Ramendra Pati Pandey, Xutao Deng, Eric Delwart, Antonio Charlys da Costa, Elcio Leal

**Affiliations:** 1Laboratório de Diversidade Viral, Instituto de Ciências Biológicas, Universidade Federal do Pará, Belem 66075-000, PA, Brazil; couto.roseane@gmail.com (R.d.S.C.); endrya.ramos@gmail.com (E.d.S.F.R.); fvillanova@gmail.com (F.V.); 2Programa de Pos-Graduação REDE Bionorte, Polo Pará, Universidade Federal do Oeste do Pará, Santarém 68040-255, PA, Brazil; uchoa_vet@yahoo.com.br; 3Laboratory of Genetics & Biodiversity, Institute of Educational Sciences, Universidade Federal do Oeste do Pará, Santarém 68040-255, PA, Brazil; luisreginaldo.ufpa@hotmail.com; 4University of Amazonia, Santarém 68040-255, PA, Brazil; fm8885785@gmail.com; 5Laboratory of Virology (LIM 52), Instituto de Medicina Tropical, Universidade de São Paulo, São Paulo 05403-000, SP, Brazil; va.morais@usp.br (V.d.S.M.); charlysbr@yahoo.com.br (A.C.d.C.); 6School of Health Sciences and Technology (SoHST), UPES, Dehradun 248007, Uttarakhand, India; ramendra.pandey@gmail.com; 7Vitalant Research Institute, San Francisco, CA 94143, USA; xutaodeng@gmail.com; 8Department Laboratory Medicine, University of California San Francisco, San Francisco, CA 94143, USA; edelwart@vitalant.org

**Keywords:** metagenomics, totivirus, bats, molossus, amazon region

## Abstract

The *Totiviridae* family of viruses has a unique genome consisting of double-stranded RNA with two open reading frames that encode the capsid protein (Cap) and the RNA-dependent RNA polymerase (RdRpol). Most virions in this family are isometric in shape, approximately 40 nm in diameter, and lack an envelope. There are five genera within this family, including *Totivirus*, *Victorivirus*, *Giardiavirus*, *Leishmaniavirus*, and *Trichomonasvirus*. While *Totivirus* and *Victorivirus* primarily infect fungi, *Giardiavirus*, *Leishmaniavirus*, and *Trichomonasvirus* infect diverse hosts, including protists, insects, and vertebrates. Recently, new totivirus-like species have been discovered in fish and plant hosts, and through metagenomic analysis, a novel totivirus-like virus (named Tianjin totivirus) has been isolated from bat guano. Interestingly, Tianjin totivirus causes cytopathic effects in insect cells but cannot grow in mammalian cells, suggesting that it infects insects consumed by insectivorous bats. In this study, we used next-generation sequencing and identified totivirus-like viruses in liver tissue from *Molossus molossus* bats in the Amazon region of Brazil. Comparative phylogenetic analysis based on the RNA-dependent RNA polymerase region revealed that the viruses identified in Molossus bats belong to two distinct phylogenetic clades, possibly comprising different genera within the *Totiviridae* family. Notably, the mean similarity between the Tianjin totivirus and the totiviruses identified in Molossus bats is less than 18%. These findings suggest that the diversity of totiviruses in bats is more extensive than previously recognized and highlight the potential for bats to serve as reservoirs for novel toti-like viruses.

## 1. Introduction

Bats are unique animals due to their extensive viral diversity, which distinguishes them from other species [1,2,3,4,5,6]. They have long been associated with many viral families and genera, such as *Paramyxoviridae*, *Filoviridae*, and *Rhabdoviridae* [7,8,9,10,11,12,13,14,15,16,17]. Their ability to fly long distances and their diverse feeding habits make it easier for them to acquire and spread viruses across remote areas and to transmit them to other species. Additionally, their social structures and behaviors contribute to virus transmission and persistence within bat populations [4,18,19,20]. However, changes in the environment, such as urbanization, agricultural intensification, and deforestation, have altered the composition and dynamics of bat communities [21]. Studies have shown an overall decline in species richness and relative abundance associated with urbanization [22]. Nevertheless, insectivorous bats tend to thrive in large urban environments [4,18,19]. Furthermore, the diversity of bat habitats can influence both microbe transmission and persistence in bat communities [2,22,23,24].

Viral metagenomics research in bats has mainly focused on North American and Eurasian bat communities [21,25,26,27]. However, studies in the Amazon region have enabled the identification of numerous viruses using conventional techniques or high-throughput sequencing [7,8,9,12,14,16,28,29]. For instance, a metagenomic study conducted in French Guiana found various RNA viruses in fecal samples of Molussus bats (also known as velvety free-tailed bats or Pallas’s mastiff bats), including some short sequences of viruses belonging to the *Totiviridae* family [30]. Another study, performed on the carcasses of deceased bats in Germany, also found short sequence of totiviruses [31]. Researchers in China were able to isolate one toti-like virus in insect cells from guano samples of the insectivorous Myotis bats [32].

Recent studies have shed light on the diverse range of insect viruses found in bat droppings through next-generation sequencing (NGS) [21,33]. NGS has facilitated the isolation and characterization of several new totivirus-like viruses that have yet to be classified by the International Committee on Taxonomy of Viruses (ICTV). The Totiviridae family is a diverse group of RNA viruses that infect both protozoa and fungi, with five identified genera to date. The family comprises 28 species, each of which contains a single molecule of double-stranded RNA (dsRNA) ranging in size from 4.6 to 7.0 kbp. The virus genome consists of two frames, with the 5′ ORF encoding the capsid protein (CP) and the 3′ ORF encoding the RNA-dependent RNA polymerase (RdRpol) gene [34].

The Totiviridae family showcases an intriguing diversity in its ability to infect a wide spectrum of hosts, spanning from fungi to protozoa, as highlighted by extensive studies [35,36,37,38,39,40,41,42]. Currently recognized genera within this family encompass *Giardiavirus*, *Leishmaniavirus*, *Totivirus*, *Victorivirus*, and *Trichomonasvirus*, each exhibiting distinct host preferences and infecting fungi or protozoa [35,43,44,45,46,47,48]. The genera *Totivirus* and *Victorivirus* primarily target fungi, while *Giardiavirus*, *Leishmaniavirus*, and *Trichomonasvirus* focus on protozoa. Recently, novel viruses sharing similar genomic structures and morphology, albeit with low similarity to existing genera, have been identified in shrimp, fish, and mosquitoes [46,49]. This discovery has led to the proposal of three new genera within the *Totiviridae* family: *Artivirus*, *Pistolvirus*, and *Tricladiviris* [38,44,47,48,50,51].

Artivirus, for instance, has demonstrated infectivity in arthropods such as the Atlantic blue crab and the mosquito *Armigere subalbatus* totivirus [52,53]. Pistolvirus, on the other hand, has been found to infect fish species like the Atlantic salmon and golden carp [38,51]. Tricladiviris exhibits its infectivity in planaria, showcasing the remarkable adaptability of totiviruses across diverse hosts. Additionally, toti-like viruses with genomic characteristics akin to Sanya orius sauteri totivirus 2 have been discovered in *Eysarcoris guttigerus*. Arboreal ants, specifically *Camponotus yamaokai*, harbor a virus that exhibits potential new totivirus traits; however, it is not phylogenetically related to the Totiviruses identified in arthropods [54].

In our research endeavors, we employed a metagenomic next-generation sequencing (NGS) approach to systematically investigate the viral landscape in the insectivorous *Molossus molossus* bats captured in the urban areas of Santarém city in northern Brazil. Our meticulous analysis yielded a total of forty-seven contigs, among which three presented near-complete genomes. These genomes exhibited an average amino acid identity of approximately 40%, showcasing a significant relatedness to a previously reported closely related totivirus-like virus.

The comprehensive exploration of the totiviruses diversity in our study not only adds valuable insights to the existing knowledge base but also underscores the intricate relationships between totiviruses and their diverse host organisms. The expanding understanding of totivirus ecology and evolution highlights the need for further research to unravel the complexities of these fascinating viruses in various ecological niches.

## 2. Materials and Methods

### 2.1. Sample Collection

We captured a total of 47 bats in Caranazal (latitude 2°26′10″ S and longitude 54°43′49″ W), a municipality in the Santarém region of Pará state in the Lower Amazon Mesoregion. These bats were identified as *Molossus molossus* (Family *Molossidae*) based on external characters. To obtain the sequences, individual bats were euthanized for sample collection. We administered xylazine hydrochloride (1 mg/kg) and ketamine hydrochloride (1–2 mg/kg) via intramuscular injection to induce anesthesia, followed by intracardiac phenobarbital (40 mg/kg) once the animals had lost consciousness. The liver samples were then collected for further analysis. Details of samples and the composition of pools are in Appendix A.

To carry out this research, approval was obtained from the Animal Use Ethics Committee of the Federal University of Western Pará (CEUA/UFOPA) under number 0220220128 and from the Biodiversity Information and Authorization System (SISBIO—18313-1) for capturing *Chiroptera*. The necropsy was performed at the Animal Morphology Laboratory of the Federal University of Western Pará, following the institutional biosafety norms.

### 2.2. Processing of Samples

To procure viral particles from liver tissue, a systematic approach was adopted, involving the preparation of organ piece pools derived from five distinct animals, each designated as F1, F2, and so forth. The tissue extraction process commenced with the maceration of the specimens in a tissue disruptor, followed by dilution in 500 µL of Hanks’ Buffered Saline Solution (HBSS). Subsequently, the samples were introduced into 2 mL tubes containing lysis matrix C (MP Biomedicals, Santa Ana, CA, USA) and subjected to homogenization using a Vortex mixer.

Following the removal of large debris, the resulting supernatants underwent filtration through 0.45 µM filters (Merck Millipore, Billerica, MA, USA) to effectively eliminate eukaryotic and bacterial cell particles. The clarified filtrates were then transferred into 1.5 mL screw cap tubes, which were subjected to centrifugation at 32,000 rpm for 1 h utilizing a Beckman Coulter Optima LE-80 ultracentrifuge equipped with a Heraeus Maximum rotor. This centrifugal step facilitated the sedimentation of viral particles, with subsequent careful removal of the supernatant. The potentially inconspicuous pellet was meticulously resuspended in 250 µL of PBS, rendering the samples amenable to nuclease enzyme treatment.

To further refine the viral extracts, the filtrates underwent treatment with DNase (concentration, 20 U/mL; Ambion, Carlsbrand, CA, USA) and RNase A (concentration, 0.1 mg/mL; Fermentas, Waltham, MA, USA) at a temperature of 37 °C for a duration of 30 min. This enzymatic intervention served to digest unprotected particle nucleic acids, enhancing the purity of the viral isolates. Additionally, the Phi29 (Φ29) polymerase enzyme was strategically employed to execute DNA circular amplification, contributing to the augmentation of the viral genetic material.

This meticulous extraction process, marked by precise steps and strategic enzymatic treatments, ensures the isolation of high-quality viral particles from liver tissue. The careful consideration given to each stage of the procedure underscores the commitment to obtaining purified and enriched viral specimens for subsequent analyses and investigations. The methodological rigor employed in this protocol sets the stage for reliable and robust downstream applications, emphasizing the significance of optimized viral extraction methodologies in advancing virological research.

### 2.3. Nucleic Acid Extraction (DNA/RNA)

After sample preparation, viral nucleic acids were extracted using the QIAamp Viral RNA Mini Kit (QIAGEN GmbH, QIAGEN Strasse 1, 40724 Hilden, Germany), which purifies RNA and DNA, and the steps were followed according to the instructions of the manufacturer.

### 2.4. Preparation of Libraries for the Illumina Platform

Library preparation was performed using the Nextera XT DNA Sample Preparation Kit (Illumina Inc., San Diego, CA, USA), following the manufacturer’s guidelines. The Agilent 2100 Bioanalyzer system and the KAPA kit were used to perform library quantification. For sequencing, samples were pooled (5 organ samples per pool). After preparing the libraries, they were sequenced on the Illumina NovaSeq-6000 platform to provide 250 bp (base pairs) paired reads (Illumina).

### 2.5. Reads Trimming and Contig Classification

The raw reads from Illumina sequencing underwent a meticulous pre-processing procedure. Initially, terminal matched sequence records were excised from both ends. Concurrently, low-quality sequences, stemming from reads shorter than 100 base pairs, were excluded [55]. The removal of adapter and primer sequences was performed with precision using VecScreen, a tool based on BLAST (Basic Local Alignment Search Tool, version BLAST 2.14.0), employing default parameters.

Subsequently to the pre-processing steps, bioinformatics analysis ensued following a well-established protocol [56]. Utilizing the bioinformatics pipeline, no reads associated with human, plant, fungal, or bacterial sequences were identified, emphasizing the specificity of the analysis. The resulting contigs underwent comparative analysis using BLASTx and BLASTn to identify similarities to viral proteins and nucleotides, respectively. The GenBank genetic sequence database (http://www.ncbi.nlm.nih.gov, accessed on 19 December 2023) served as the reference for this comparison.

In addition to the comparative analysis of the predicted gene sequences via the BLASTx online program, known for its protein alignment capabilities using DNA sequences, the most promising outcomes from the BLAST searches were meticulously selected. To minimize the potential for random matches, E values (e-value) were defined for each search. Based on the best result, the sequences were chosen for further alignment.

For confirmation and additional classification, reads and/or contigs underwent alignment against a viral protein database (obtained from https://ftp.ncbi.nlm.nih.gov/refseq/release/viral/, accessed 10 January 2023) using DIAMOND software version 2.1.8 [57]. All sequences generated in this study were deposited in GenBank with the accession numbers OR069303-OR069349.

All sequences included in this study exhibited similarity to reference sequences from GenBank classified in the *Totiviridae* family. Complete or almost complete genomes were aligned (using MAFFT version 7.520) [58], with further adjustments and editions performed using the Ugene tool kit version 4.6 [59].

### 2.6. Genome Annotation

The RdRpol domains and motifs were predicted using InterProScan (https://www.ebi.ac.uk/interpro/search/sequence, accessed 10 January 2023) and Motif Finder (https://www.genome.jp/tools/motif, accessed 12 January 2023), respectively.

### 2.7. Genetic Distances

The genetic distance and its standard error were calculated using the composite model of maximum likelihood plus gamma correction and bootstrapping with 1000 replications. Distances were calculated using the MEGA X software package, Version X [60]. To estimate the similarity of the sequences, a paired method was used, implemented in the SDT program [61]. The initial realignment of sequences involved penalizing gaps, a process carried out with the MUSCLE algorithm [62]. Following the calculation of identity scores for each pair of sequences (pairwise scores), the NEIGHBOR component of PHYLIP [63] was employed to construct a tree. This rooted neighbor-joining phylogenetic tree organizes all sequences based on their likely degrees of evolutionary relatedness. The outcomes are illustrated in a graphical interface through a frequency distribution of paired identities, forming an identity matrix.

### 2.8. Phylogenetic Analysis

Phylogenetic trees were meticulously constructed utilizing the maximum likelihood approach, and robust branching support was diligently estimated through a bootstrap test comprising 1000 iterations, employing the IQ-Tree tool [64]. Subsequently, the maximum clade credibility tree was derived employing a Bayesian coalescent approach, skillfully implemented within the Beast v1.10.4 software [65] (accessible at https://github.com/beast-dev/beast-mcmc, accessed 10 January 2023). In this analytical framework, a fixed clock and a constant rate of population size were assumed, along with the consideration that the evolutionary rate of a specific site in a gene-sequence alignment remained constant throughout evolution, adhering to the concept of homotachy.

The evolutionary model employed was WAG + I + G, with the number of gamma categories set to 4. The simulation runs were initiated using a randomly generated starting tree and an extensive chain length of 100,000,000, incorporating echo state-to-screen evaluations every 10,000 iterations, log parameter assessments every 10,000 iterations, and a burn-in phase of 10%. In order to guarantee convergence and ascertain an effective sample size (ESS) surpassing 200, a meticulous examination of the outcomes was conducted utilizing Tracer v1.7.1 [66] (accessible at http://tree.bio.ed.ac.uk/software/tracer, accessed 10 January 2023). This thorough analytical methodology ensures the reliability and robustness of the phylogenetic inferences derived from the intricate computational processes involved in tree construction and evolutionary modeling.

## 3. Results

### 3.1. Contigs Quality and BlastX Similarities

We successfully retrieved forty-seven contigs from three different pools (F1, F3, and F6), which were identified as totiviruses based on blast searches. The contigs’ quality, sequence size, and blast similarities are summarized in Appendix A. The general quality of contigs was high due to the large number of reads in the assembly of each contig. A BlastX comparative analysis showed that contigs identified in liver samples of Molossus bats had low amino acid identity in the NCBI database (mean 48%), with coverage ranging from 15% to 97% with their best-hit reference sequence. Three of the contigs (F1_001, F1_002, and F1_003) represent the near full-length genome of totiviruses.

### 3.2. Genome Annotation of Totiviruses Identified in Bats 

We have discovered three nearly complete genomes of totiviruses in liver samples collected from Molossus bats, specifically F1_001, F1_002, and F1_003. Using a BlastX search, we found that these two sequences share a degree of amino acid similarity with totiviruses previously identified in the helminth *Schistocephalus solidus*. To further explore this similarity, we compared the genome annotations of all these sequences (Figure 1). It is worth noting that the reference sequences MN803435 and MN803437, identified in *Schistocephalus solidus* [67], are related, but they have distinct genome maps. MN803437 has an additional open reading frame (orf), and the proteins have different sizes. For instance, the polymerase in MN803437 has 840 amino acids, while in MN803435, it has 953 amino acids. Both F1_001 and F1_003 contain three open reading frames (ORFs), with F1_003 having a notably short hypothetical polymerase sequence (only 486 amino acids). An important characteristic of these sequences is the presence of long non-coding intergenic regions, such as the 538 bp region found between the second complete genome. F1_002 has two hypothetical proteins: the first has 1456 residues, and the second (probably RdRpol) has 1042 residues.

### 3.3. RNA-Dependent RNA-Polymerase of Totiviruses

Due to the limited similarities between our contigs and the reference sequences, we opted to analyze the motifs of the RdRpol region of totiviruses. To achieve this, we utilized the cognate viruses identified through the Blast search as well as additional contigs found in Molossus bats (Figure 2). To illustrate our findings, we have included the motifs of RdRpol from three contigs that belong to different groups of totiviruses (as determined by our phylogenetic analysis). In addition to these motifs, we have also shown the motifs of their respective best hits. It is important to note that even though the RdRpol sequences identified in this study are highly divergent (more than 50%), all of the motifs of the polymerase are present. Typically, cognate sequences share the same pattern, with one exception being the sequence F1_001, which possesses an additional motif (indicated by an arrow in Figure 2a) when compared to its best-hit reference, QJD26160, identified in *Schistocephalus solidus*.

### 3.4. Phylogenetic Tree of RdRpol of Totiviruses

Phylogenetic inference was performed using amino acid sequences of the RdRpol region to classify the sequences generated in this study. To this end, we utilized 53 selected reference sequences from the Totiviridae family. The resulting phylogenetic tree (Figure 3) was generated using a Coalescent Bayesian approach. The tree accurately depicts the major genera of totiviruses, including *Giardiavirus*, *Trichomonavirus*, *Leishmaniairus*, *Victorivirus*, and *Totivirus*. Furthermore, the recently proposed genera *Artivirus*, *Pistolvirus*, and *Tricladivirus* are also identified in this tree. The sequences from this study were grouped into two distinct phylogenetic clades in the tree (indicated in blue color in the tree of Figure 3). Most sequences (i.e., F1_001, F1_006, F1_032, F1_066, F1_046, F1_052, F1_010, F1_012, F1_051, F1_042, F1_079, F1_045, F1_058, F1_043, F1_044, F1_036, F1_003, F1_062, F1_021, F1_007, F1_048, F1_081, F1_064, F1_008, and F1_002) clustered in a clade (here named Platyhelminthes) that includes the sequences QJD26156 and QJD26160 identified previously in the platyhelminthes *Schistocephalus solidus* in the United States in 2018. The sequences F1_071, F1_047, F1_067, and F1_080 form a cluster (named Insecta) with the reference UHR49805 that was identified in the insect *Eysarcoris guttigerus* in China in 2017. 

### 3.5. RdRpol Amino Acid Identity 

Our phylogenetic analysis revealed that the sequences in the Platyhelminthes clade generated in this study could be categorized into two subclades. To further examine the level of identity of the RdRpol of these sequences, we created a subtree containing the Platyhelminthes clade and used it to illustrate the level of identity (Figure 4). The amino acid identity between F1_008 and its sister sequence, F1_002, was found to be 83%. The mean identity among sequences in the clade composed of F1_001, F1_006, F1_032, F1_066, F1_046, F1_052, F1_010, F1_012, F1_051, F1_042, F1_079, F1_045, F1_058, F1_043, F1_044, F1_036, F1_003, F1_062, F1_021, F1_007, F1_048, F1_081, and F1_064 is 80%. In contrast, the identity between the sequences identified in *Schistocephalus solidus* is only 59%. On the other hand, the mean identity of sequences identified in *Schistocephalus solidus* with F1_001 and F1_002 is 35%. There is a 40% level of identity between F1_001 and F2_002. Appendix A illustrates the identities of every pair of sequences generated during the present study.

## 4. Discussion

Viral surveillance in bats has undergone extensive scrutiny in southern Brazil, with a notable emphasis on the exploration of bat viruses in this region. However, there is a conspicuous gap in studies examining bat viruses in other geographical areas. Previous comprehensive investigations in Brazil have successfully identified a diverse array of viruses in bats, including Flavivirus, Coronavirus, Arenavirus, Paramyxovirus, Adenovirus, Papillomavirus, and Parvovirus. These findings were particularly notable in bats such as *Molossus molossus*, *Artibeus lituratus*, and *Sturnira lilium* [7,9,14,16,20,28,29,68,69].

The advent of next-generation sequencing technologies has revolutionized the landscape of viral detection in bats. This technological leap has led to the identification of numerous viruses in bats, as evidenced by a plethora of studies [6,25,26,27,70]. Despite these advancements, reports on totiviruses in bats remain relatively scarce [27,32]. A recent breakthrough emerged when two contigs, measuring 2059 bp and 627 bp, of a totivirus-like virus were uncovered in the insectivorous bat *Nyctalus noctula* during a study focused on bat carcasses [31]. Additionally, short reads of totivirus were detected in the feces of *Molossus molossus* in French Guiana, as reported in another insightful study [30].

It is imperative to highlight a crucial aspect inherent in metagenomics studies: the inherent challenge of pinpointing the exact viral host due to the nature of next-generation sequencing approaches [33]. This challenge persists even in the context of totiviruses identified in bats, where hosts may range from ecto- or endoparasites to the potential food sources consumed by these animals [29,30,31,32].

A noteworthy case in point is the singular totivirus-like virus isolated in insect cells, aptly named Tianjin totivirus, which was detected in the guano of *Myotis* sp. bats [32]. It is postulated that this virus likely infects insects that form a significant part of the diet of insectivorous bats [29,30,31,32,33]. This intriguing revelation opens up avenues for further exploration into the intricate dynamics of host–virus interactions within bat populations and their associated ecosystems.

As we navigate the complexities of viral surveillance in bats, these insights underscore the need for continued research to unravel the mysteries surrounding totiviruses and their diverse interactions with bat hosts. The evolving landscape of next-generation sequencing technologies provides a promising platform for future discoveries, pushing the boundaries of our understanding of the intricate relationships between bats and the viruses they harbor. In conclusion, while the current knowledge base offers valuable insights, the vast realm of undiscovered viral diversity in bats beckons researchers to delve deeper into unexplored territories, fostering a more comprehensive understanding of the dynamics between bats and the viruses that inhabit them.

In our investigation of liver samples from Molossus bats, we unveiled a total of 47 contigs, each displaying varying degrees of amino acid similarity with totivirus-like sequences previously identified in a diverse range of hosts. The phylogenetic analysis conducted on these sequences brought to light twenty-five RdRpol sequences closely associated with the platyhelminthes *Schistocephalus solidus*, a species detected in the United States in 2018. Furthermore, our scrutiny identified an additional four RdRpol sequences linked to a totivirus reference discovered in the insect *Eysarcoris guttigerus* in China in 2017.

An essential point to underscore is the amino acid identity of RdRpol between our newly discovered sequences and the totivirus-like reference, which falls below 50%. This discrepancy indicates a significant genetic divergence between these viruses, suggesting potential evolutionary divergence. Interestingly, the Tianjin totivirus, isolated from guano bats, demonstrates less than 18% amino acid identity in the RdRpol region compared to the totivirus-like sequences identified in Molossus bats. This stark contrast strongly implies the ancient nature of these viruses and underscores their possession of host-specific characteristics.

This current study employed next-generation sequencing techniques to explore totivirus-like viruses present in the liver tissue of Molossus bats residing in the Amazon region of Brazil. Through a meticulous comparative phylogenetic analysis focusing on the RNA-dependent RNA polymerase region, we made the intriguing observation that the viruses identified in Molossus bats fall into two distinct phylogenetic clades. This finding hints at the possibility that these viruses may represent different genera within the *Totiviridae* family.

The revelation of these distinct phylogenetic clades raises questions about the potential diversity and adaptations within totivirus-like viruses in Molossus bats. The implications extend to the possibility that these viruses might be associated with different hosts or possess unique characteristics that warrant further investigation.

Moreover, the presence of totivirus-like viruses in Molossus bats prompts speculation about the targeted organism for replication. Given the viruses’ identification in liver tissues, it suggests a potential association with an organism that infects the liver of these bats. This opens avenues for future research to delve into the specifics of the host–virus interactions and the implications for the overall ecology and health of Molossus bat populations in the Amazon region.

## 5. Conclusion

Our expanded exploration into the genetic diversity of totivirus-like viruses in Molossus bats not only highlights their intriguing evolutionary patterns and divergence but also prompts further inquiries into their potential ecological roles and impacts on host populations. The results of this study will help direct future investigations into the intricate relationships that these viruses have with their host organisms.

## Figures and Tables

**Figure 1 microorganisms-12-00206-f001:**
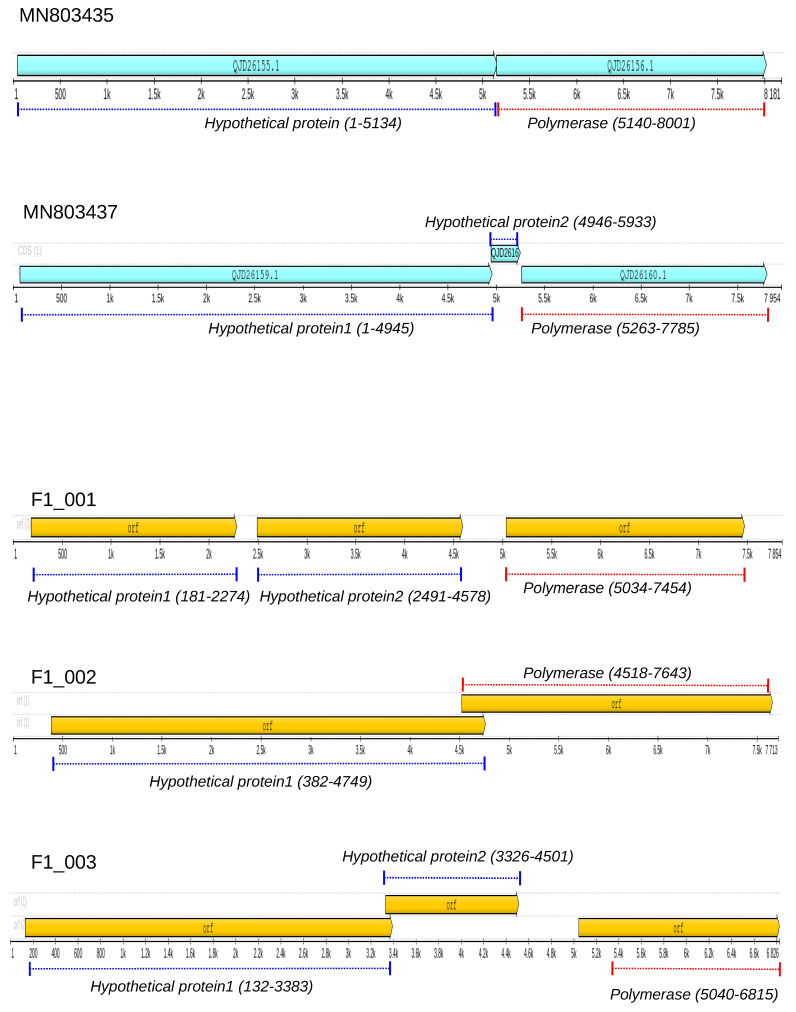
Genome map of totiviruses. Horizontal diagrams represent the location of hypothetical open reading frames (ORF) in the genome of totiviruses. Upper panels indicate the ORFs (cyan bars) detected in the genomes of sequences MN803435 and MN803437, identified in *Schistocephalus solidus*. Lower panels show the ORFs (orange bars) detected in the genomes of F1_001, F1_002, and F1_003 identified in Molossus bats.

**Figure 2 microorganisms-12-00206-f002:**
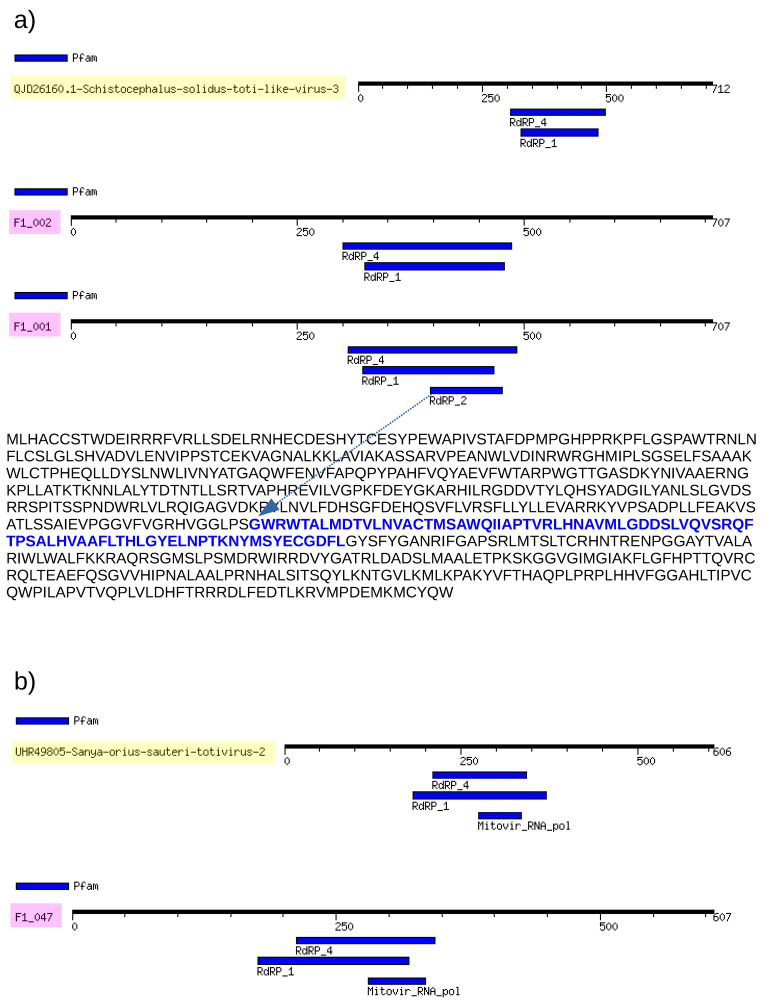
RdRpol profiles of totiviruses. The translated polymerase sequence of totiviruses is represented by horizontal bars that show the location of RdRpol motifs. (**a**) The polymerase of F1_001 and F1_002 were compared to their best-hit reference QJD26160 (identified in *Schistocephalus solidus*), and the identified motifs are shown. In addition, F1_001 has an extra motif that is highlighted in blue and indicated by an arrow in this figure. (**b**) In sequence F1_047, the motifs are compared to its cognate best-hit reference UHR49805, which was identified in the herbivorous insect *Eysarcoris guttigerus* that feeds on plants.

**Figure 3 microorganisms-12-00206-f003:**
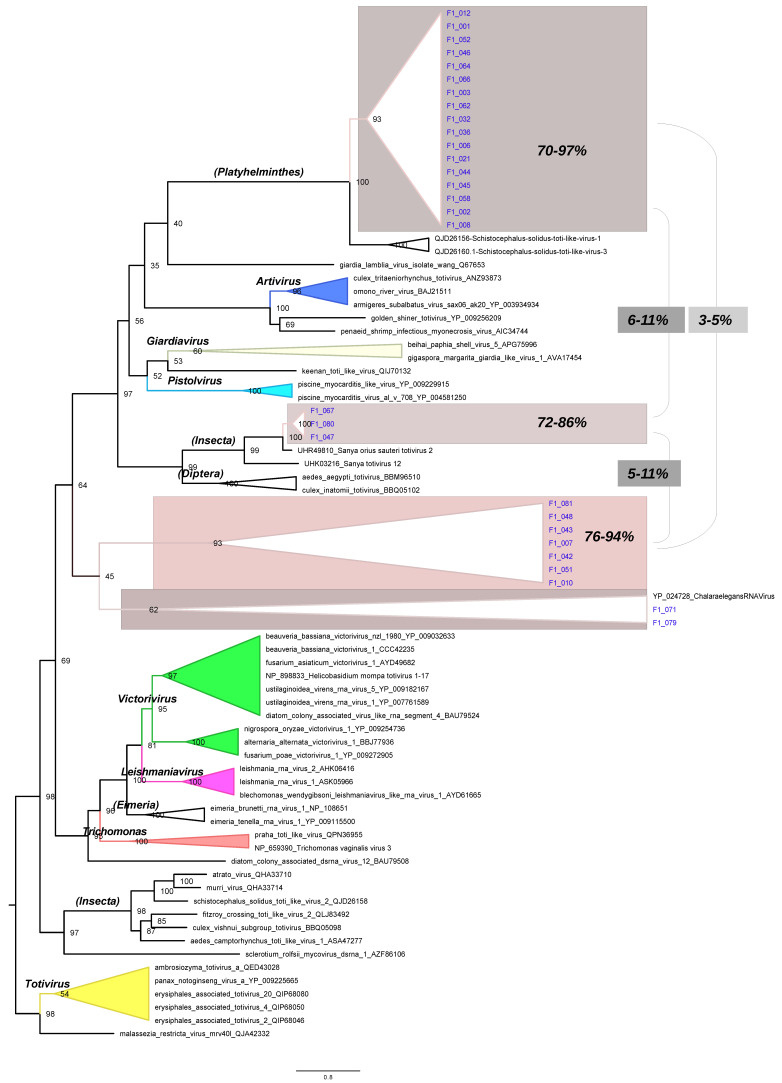
Phylogenetic tree of RdRpol of totiviruses. The translated RdRpol sequences were used to infer the tree, and branch support is based on bootstrap test, which is indicated by colors according to the scale in the upper part of the figure. Clades with high branch support are represented by triangles in the tree. The horizontal bar shows the scale of the tree in substitutions per site. Sequences identified in this study are indicated in blue color in the tree. The names above the branches indicate the main totivirus genera as per the International Committee on Taxonomy of Viruses (ICTV). Names inside parentheses, such as Platyhelminthes and Insecta, are not genera; rather, they represent clades identified in this study. Colored triangles represent different genera. Numbers indicate the amino acid identity among sequences. Clades that contain sequences generated in this study were highlighted.

**Figure 4 microorganisms-12-00206-f004:**
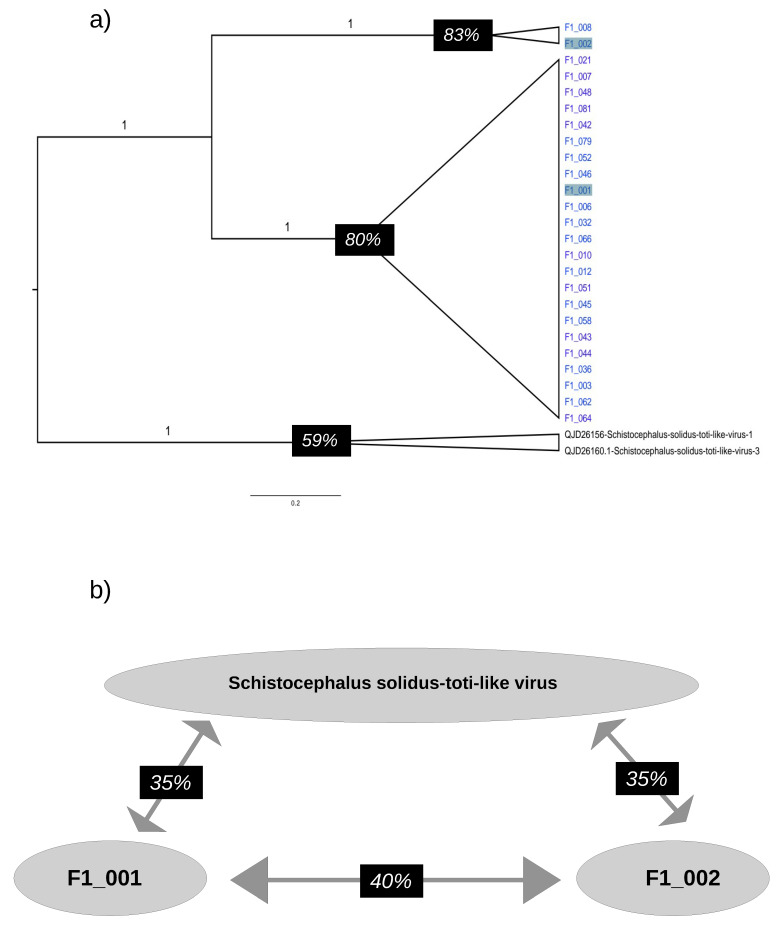
Amino acid identity in the RdRpol of totiviruses. (**a**) Subtree depicting the phylogenetic clade Platyhelminthes. Values above branches indicate the posterior probability. The identity within each clade is indicated by black rectangles in the tree. (**b**) Identity of sequences identified in Schistocephalus solidus with F1_001 and F1_002 (these sequences are highlighted in the tree).

## Data Availability

The raw data presented in this study are available on request from the corresponding author.

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
