# Peer review of "Metagenomic of Liver Tissue Identified at Least Two Genera of Totivirus-like Viruses in Molossus molossus Bats"

_microorganisms, 2024, doi:10.3390/microorganisms12010206_

Round 1

Reviewer 1 Report

Comments and Suggestions for Authors In this study, the authors used next-generation sequencing and identified two totivirus-like viruses in liver tissue from Molossus molossus bats in the Amazon region of Brazil. Only two viral contigs analysis was not enough to support its publishing. More viral information should be explored.  1. If possible, please amplify the whole genome of these two totivirus complete genome and get more characters of bat-related totiviruses. 2. How is the epidemiology of these totiviruses in the collected liver samples, please provide more results. 3. Could the authors study  further about the host of these totiviruses?  Does it infect bat or not? How is the pathology of these bat livers?

Author Response

In this study, the authors used next-generation sequencing and identified two totivirus-like viruses in liver tissue from Molossus molossus bats in the Amazon region of Brazil. Only two viral contigs analysis was not enough to support its publishing. More viral information should be explored.

RESP: We identified forty-seven contigs of Totilikevirus from three different pools (F1, F3, and F6).

1. If possible, please amplify the whole genome of these two totivirus complete genome and get more characters of bat-related totiviruses.

RESP: Some of these contigs are indeed full-length genomes. Please check the details of these contigs in TableS2.

2. How is the epidemiology of these totiviruses in the collected liver samples, please provide more results.

RESP: Visual inspection by trained veterinarians revealed that most animals were healthy. We found three animals with moderate hepatomegaly.

3. Could the authors study further about the host of these totiviruses? Does it infect bat or not? How is the pathology of these bat livers?

RESP: As pointed out in our, hosts of totiviruses found in metagenomic of bats, can be ecto- or endoparasites, or even the food source consumed by these animals. The only totivirus-like virus that has been isolated in insect cells, named Tianjin totivirus, was detected in the guano of Myotis bats [see reference 32 of our manuscript].

Reviewer 2 Report

Comments and Suggestions for Authors

In this work, authors used next-generation sequencing and identified three totivirus-like viruses in liver tissue from Molossus molossus bats in the Amazon region of Brazil. These findings suggest that the diversity of totiviruses in bats is more extensive. There is no doubt that this study provided a lot of valuable information for totiviruses. However, some should be revised.

1)line 62-the reference number is repeated.

2)line 72,84,85, Totiviridae may be italic.

3)line 93, Totiviruses is totiviruses.

4) Authors should add a reverse-transcription polymerase chain reaction (RT-PCR) test for validation of virus-like contigs.

5) The full-length sequence of each totivirus-like virus should be obtained by using RT-PCR and rapid amplification of cDNA ends (RACE) protocol.

6)line 117, authors mentioned that “To extract viral particles from liver tissue” however, in the Results, no described about the viral particles. Why?

Author Response

In this work, authors used next-generation sequencing and identified three totivirus-like viruses in liver tissue from Molossus molossus bats in the Amazon region of Brazil. These findings suggest that the diversity of totiviruses in bats is more extensive. There is no doubt that this study provided a lot of valuable information for totiviruses. However, some should be revised.

1)line 62-the reference number is repeated.

RESP: This was corrected in the new version of the manuscript

2)line 72,84,85, Totiviridae may be italic.

RESP: This was corrected in the new version of the manuscript

3)line 93, Totiviruses is totiviruses.

RESP: This was corrected in the new version of the manuscript

4) Authors should add a reverse-transcription polymerase chain reaction (RT-PCR) test for validation of virus-like contigs.

RESP: We did PCR to confirm the presence of totivirus-like in the pools used for NG-sequencing

5) The full-length sequence of each totivirus-like virus should be obtained by using RT-PCR and rapid amplification of cDNA ends (RACE) protocol.

RESP: Since most contigs were near-complete genomes we decided to perform RACE besides we had a shortage of reagents

6)line 117, authors mentioned that “To extract viral particles from liver tissue” however, in the Results, no described about the viral particles. Why?

RESP: This is an error, we changed the sentence to “viral RNA”

Reviewer 3 Report

Comments and Suggestions for Authors

Couto et al, have described next- generation sequencing approach to identify Totivirus -like virus from Molossus molossus bats.   to express VZV glycoprotein E.

This is an interesting premise, and identification of the viruses in relation to bats as a host  has been picturesque, using simple but effective experiments. The results are intriguing and potentially quite exciting. However, there are a

few important points that need to be addressed.

1)    A high-resolution image of Phylogenetic tree of Totivirus based on RdRpol (Figure 3) should be provided.

2)    In the legend of figure 2 please define the sub section a) and b) of the figures.

3)    Please provide the information if the isolated virus like particles were used for infection in the insects or bat cells for virus isolation. 

4)    [ Lines 72-73] Relevant citations should be added.

5)     [ Lines 324-325] Relevant citations should be added.

6)    [ Lines 327] Relevant citations should be added.

7)     [ Lines 329] Relevant citations should be added.

Comments on the Quality of English Language

Minor editing of English language is required. Please do check for the punctuation's. 

Author Response

Couto et al, have described next- generation sequencing approach to identify Totivirus -like virus from Molossus molossus bats. to express VZV glycoprotein E.

This is an interesting premise, and identification of the viruses in relation to bats as a host has been picturesque, using simple but effective experiments. The results are intriguing and potentially quite exciting. However, there are a few important points that need to be addressed.

1) A high-resolution image of Phylogenetic tree of Totivirus based on RdRpol (Figure 3) should be provided.

Resp:We did increase the resolution of figure 3 in the new version of the manuscript

2) In the legend of figure 2 please define the sub section a) and b) of the figures.

Resp We did change the legend of Figure 2 to clarify the sub-sections a and b

3) Please provide the information if the isolated virus like particles were used for infection in the insects or bat cells for virus isolation.

Resp:

4) [ Lines 72-73] Relevant citations should be added.

Resp: We did add a reference to this sentence

5) [ Lines 324-325] Relevant citations should be added.

Resp:We did add a reference to this sentence

6) [ Lines 327] Relevant citations should be added.

Resp: We did add a reference to this sentence

7) [ Lines 329] Relevant citations should be added.

Resp: We did add a reference to this sentence

Round 2

Reviewer 1 Report

Comments and Suggestions for Authors

The authors answered most of the questions. However, the toti-like viral contigs was detected in bat liver, not same as Tianjin totivirus, it should be originated from bat and infect mammal, please analyze more about this point.

 Minor:

1. Line 147, please double check "32,000 rpm for 5 minutes", ultra-high speed centrifigation usually should be more than 1 hour.

Author Response

The authors answered most of the questions. However, the toti-like viral contigs was detected in bat liver, not same as Tianjin totivirus, it should be originated from bat and infect mammal, please analyze more about this point.

We strongly agree that there is much more to explore before asserting that the totiviruses we discovered infect bats. While we didn't find reads of helminths, and visual inspections of the liver didn't reveal any lesions, we cannot rule out the possibility that these viruses are infecting some parasitic organisms within these bats.

 Minor:

1. Line 147, please double check "32,000 rpm for 5 minutes", ultra-high speed centrifigation usually should be more than 1 hour.

You are right the correct time is one hour, we changed the text in the manuscript.